# Anti-Atherogenic Actions of the Lab4b Consortium of Probiotics In Vitro

**DOI:** 10.3390/ijms24043639

**Published:** 2023-02-11

**Authors:** Victoria L. O’Morain, Jing Chen, Sue F. Plummer, Daryn R. Michael, Dipak P. Ramji

**Affiliations:** 1Cardiff School of Biosciences, Cardiff University, Sir Martin Evans Building, Museum Avenue, Cardiff CF10 3AX, UK; 2Cultech Limited, Unit 2 Christchurch Road, Baglan Industrial Park, Port Talbot SA12 7BZ, UK

**Keywords:** atherosclerosis, foam cells, macrophages, gene expression, inflammation, probiotics, smooth muscle cells

## Abstract

Probiotic bacteria have many protective effects against inflammatory disorders, though the mechanisms underlying their actions are poorly understood. The Lab4b consortium of probiotics contains four strains of lactic acid bacteria and bifidobacteria that are reflective of the gut of newborn babies and infants. The effect of Lab4b on atherosclerosis, an inflammatory disorder of the vasculature, has not yet been determined and was investigated on key processes associated with this disease in human monocytes/macrophages and vascular smooth muscle cells in vitro. The Lab4b conditioned medium (CM) attenuated chemokine-driven monocytic migration, monocyte/macrophage proliferation, uptake of modified LDL and macropinocytosis in macrophages together with the proliferation and platelet-derived growth factor-induced migration of vascular smooth muscle cells. The Lab4b CM also induced phagocytosis in macrophages and cholesterol efflux from macrophage-derived foam cells. The effect of Lab4b CM on macrophage foam cell formation was associated with a decrease in the expression of several key genes implicated in the uptake of modified LDL and induced expression of those involved in cholesterol efflux. These studies reveal, for the first time, several anti-atherogenic actions of Lab4b and strongly implicate further studies in mouse models of the disease in vivo and in clinical trials.

## 1. Introduction

Atherosclerosis is an inflammatory disease associated with the build-up of lipids and cellular debris in the walls of arteries, and the underlying cause of cardiovascular diseases such as myocardial infarction, cerebrovascular accidents, and peripheral vascular disease [1,2,3]. Current therapies against atherosclerosis are associated with considerable residual risk for the disease together with various adverse side effects in some individuals [2,3,4]. In addition, many promising leads from drug discovery and other screening programs have failed because of side effects and off-target actions [2,3,4]. Nutraceuticals, such as probiotic bacteria, have an excellent safety profile and offer substantial promise in the prevention and treatment of atherosclerosis and as adds-on with current pharmaceutical drugs, such as statins [1,3,4]. Unfortunately, the mechanisms underlying the protective actions of nutraceuticals are poorly understood and further studies are required, as they can form the foundations for clinical trials.

In relation to probiotic bacteria, our previous study showed that Lab4P [Lab4 consortium of probiotics composed of *Lactobacillus acidophilus* CUL21 (NCIMB 30156) and CUL60 (NCIMB 30157), *Bifidobacterium bifidum* CUL20 (NCIMB 30153) and *Bifidobacterium animalis* subsp. *lactis* CUL34 (NCIMB 30172) combined with *Lactobacillus plantarum* CUL66], reduced atherosclerosis-associated risk factors, such as plasma total cholesterol levels and diet-induced weight gain, in C57BL/6J mice fed a high fat diet (HFD) [5]. More recently, we have shown that Lab4P causes plaque stabilisation and attenuates atherosclerosis development and plaque inflammation in LDL receptor-deficient mice fed a HFD [6]. In addition, the use of conditioned medium (CM) from these probiotic bacteria demonstrated beneficial regulation of several atherosclerosis-associated processes in vitro, such as inhibition of chemokine-driven monocytic migration, proliferation of monocytes and macrophages, foam cell formation and associated changes in the expression of key genes, and proliferation and migration of vascular smooth muscle cells (VSMC) together with stimulation of cholesterol efflux from macrophage foam cells [6].

The Lab4b consortium of probiotics contains four strains of lactic acid bacteria and bifidobacteria [*Lactobacillus salivarius* CUL61 (NCIMB 30211), *Lactobacillus paracasei* CUL08 (NCIMB 30154), *Bifidobacterium bifidum* CUL20 (NCIMB 30153), and *Bifidobacterium animalis* subsp. *lactis* CUL34 (NCIMB 30172)] and has been specifically designed to reflect the gut of newborn babies and infants [7,8]. CM from these probiotic bacteria have previously been shown to exert neuroprotective activities in vitro [9]. The effects of Lab4b CM on atherosclerosis have not been determined. The objective of this study was, therefore, to investigate the actions of Lab4b CM on monocytes/macrophages and VSMC, two key cell types implicated in atherosclerosis [1], in vitro, together with the underlying molecular mechanisms. Lab4b CM exerted several anti-atherogenic activities, including attenuation of chemokine-driven monocytic migration, macrophage foam cell formation, monocyte/macrophage proliferation, and platelet-derived growth factor (PDGF)-mediated migration of VSMC.

## 2. Results

### 2.1. Lab4b CM Has No Detrimental Effect on Cell Viability of Human Macrophages and VSMC

The human THP-1 cell line is used extensively for the investigation of monocyte/macrophage processes in inflammation and atherosclerosis because of conservation of responses with primary cultures and in vivo [6,10,11].They were therefore used for analysis of the actions of Lab4b CM with key findings validated in primary cultures of human monocyte-derived macrophages (HMDM). Primary cultures of human aortic smooth muscle cells (HASMC) were used as a model for the effects of Lab4b CM on this cell type in relation to atherosclerosis [6,11]. The highest concentration of Lab4b CM used in our studies was 1.5 μg/mL; therefore, the effect of this concentration on cell viability was determined. No detrimental reduction in cell viability by Lab4b CM was seen in THP-1 macrophages, HMDM, and HASMC (Appendix A).

### 2.2. Lab4b Attenuates Chemokine-Driven Monocytic Migration

Monocytic migration induced by key chemokines, such as monocyte chemotactic protein-1 (MCP-1), is a key early event in the pathogenesis of atherosclerosis [3]. The effect of three different concentrations of Lab4b CM on MCP-1-driven monocytic migration was determined and, as expected, inclusion of this chemokine induced the migration of THP-1 monocytes (*p* ≤ 0.001) (Figure 1). In addition, this MCP-1-induced migration of THP-1 monocytes was attenuated by all three concentrations of Lab4b CM in a dose-dependent manner (*p* = 0.005 for 0.5 μg/mL and *p* ≤ 0.001 for 1 μg/mL and 1.5 μg/mL).

### 2.3. Lab4b CM Has Beneficial Actions on Several Parameters Associated with Macrophage Foam Cell Formation

Macrophage foam cell formation is a critical event in atherogenesis and involves production of reactive oxygen species (ROS) that is then involved in the oxidation of LDL, uptake of lipoproteins/modified lipoproteins by macropinocytosis, phagocytosis, and receptor-mediated endocytosis, and efflux of cholesterol from foam cells [12]. The effects of Lab4b CM on parameters related to macrophage foam cell formation were therefore determined.

The effects of three different concentrations of Lab4b CM on tert-butyl hydroperoxide (TBHP)-induced ROS production were determined in both THP-1 monocytes and macrophages. As expected, the inclusion of TBHP induced ROS production in both THP-1 monocytes and macrophages (*p* ≤ 0.001) (Figure 2). In addition, this TBHP-induced ROS production in THP-1 monocytes was significantly increased by Lab4b CM at 0.5 μg/mL (*p* = 0.002) and showed a non-significant trend towards increase with 1 μg/mL (*p* = 0.056) (Figure 2A). Similarly, the TBHP-induced ROS production in macrophages was significantly increased by all three concentrations of Lab4b CM (*p* ≤ 0.001 for both 0.5 μg/mL and 1.5 μg/mL, and *p* = 0.002 for 1 μg/mL) (Figure 2B).

Macropinocytosis in THP-1 macrophages, as determined by the uptake of the Lucifer yellow dye, was attenuated by all three concentrations of Lab4b CM (*p* ≤ 0.001 in all cases) (Figure 3A). The maximum reduction was seen at a concentration of 1.5 μg/mL; therefore, this was used for all subsequent assays. Lab4b CM at this concentration attenuated Dil-oxidised LDL (Dil-oxLDL) uptake in THP-1 macrophages (*p* ≤ 0.001; Figure 3B). A similarly significant reduction was observed when the experiment was repeated in primary cultures of HMDM (*p* = 0.004; Figure 3C), thereby ruling out the possibility that the results were because of the transformed nature of THP-1 cell line. In contrast to these actions, Lab4b CM significantly induced phagocytosis in THP-1 macrophages (*p* = 0.004; Figure 3D) and cholesterol efflux from THP-1 macrophage-derived foam cells (*p* ≤ 0.001; Figure 3E).

Scavenger receptor CD36 plays a key role in the receptor-mediated uptake of modified LDL, and Lab4b CM significantly decreased its expression in both THP-1 macrophages (*p* ≤ 0.001; Figure 4A) and HMDM (*p* ≤ 0.001; Figure 4B). Lab4b CM also significantly decreased the expression of two other key genes implicated in the uptake of modified LDL in THP-1 macrophages: SRA (*p* ≤ 0.001; Figure 4C) and LPL (*p* ≤ 0.001; Figure 4D). The effect of Lab4b CM on five key genes implicated in cholesterol efflux [ATP-binding cassette transporter (ABC)A1, ABCG1, liver-X-receptor (LXR)-α, LXR-β, and ApoE] were also determined in THP-1 macrophage-derived foam cells. Compared to the vehicle control, Lab4b CM significantly induced the expression of ABCA1 (*p* ≤ 0.001; Figure 4E), ABCG1 (*p* ≤ 0.001; Figure 4F), LXR-α (*p* ≤ 0.001; Figure 4G), and LXR-β (*p* ≤ 0.001; Figure 4H) without affecting that of ApoE (Figure 4I).

### 2.4. The Effects of Lab4b CM on Monocyte/Macrophage Proliferation

The effects of Lab4b CM on proliferation of monocytes and macrophages, which is increasingly being identified as an important contributor to their burden in atherosclerotic plaques [13], was next determined. For THP-1 monocytes, cell numbers were counted at days 2, 5, and 7 following treatments of the cells with vehicle or Lab4b CM. At day 2, Lab4b CM produced a non-significant reduction in cell numbers compared to the vehicle control (Figure 5A). However, a significant reduction in cell numbers was seen at days 5 and 7 compared to the vehicle control (*p* = 0.002 for day 5, *p* = 0.010 for day 7, respectively; Figure 5B,C). A similar outcome was seen when THP-1 monocytes treated with either vehicle or Lab4b CM were counted daily for 7 days, and proliferation determined as a percentage change in cell number by linear regression analysis (Appendix A).

For THP-1 macrophages, the cells were treated with either vehicle or Lab4b CM for 48 h and cellular proliferation was assessed by crystal violet staining. To control for any changes in cell viability, the LDH assay was performed simultaneously. Lab4b CM significantly attenuated proliferation (*p* ≤ 0.001) (Figure 5D) without a detrimental effect on cell viability (Figure 5E). To confirm such an effect, the rate of cell proliferation was assessed by monitoring 5-bromo uridine (BrdU) incorporation after 48 h of treatment with vehicle or Lab4b CM. The Lab4b CM significantly attenuated the rate of proliferation of THP-1 macrophages (*p* = 0.025; Figure 5F).

### 2.5. Lab4b CM Attenuates HASMC Proliferation and Platelet-Derived Growth Factor (PDGF)-Induced Migration

HASMCs were treated with either vehicle or Lab4b CM and proliferation was assessed by crystal violet staining over a 7-day period. Linear regression analysis showed that Lab4b CM produced a non-significant reduction in proliferation compared to day 0 (Appendix A). To evaluate the effect of Lab4b CM on proliferation compared to the vehicle control, the crystal violet and LDH assays were carried out on day 7 of treatment. Lab4b CM produced a significant reduction in cell numbers (*p* = 0.008; Figure 6A) without affecting cell viability (Figure 6B). To further analyse the effect of Lab4b CM on rate of proliferation, BrdU incorporation assay was carried out after 2, 5, and 7 days. Compared to the vehicle control, a significant reduction in rate of proliferation was seen at day 2 (*p* < 0.001; Figure 6C) but not days 5 and 7 (data not shown). Lab4b CM also significantly attenuated PDGF-mediated migration of HASMCs in vitro (*p* < 0.001; Figure 6D).

## 3. Discussion

The benefits of probiotics have been proposed for many pathologies, including cancer, cardiovascular diseases (e.g., coronary heart disease, stroke) and neurodegenerative disorders (e.g., Azheimer’s disease, mild cognitive impairment, Parkinson’s disease) [14,15,16,17]. Unfortunately, the molecular mechanisms underlying such beneficial actions remain poorly understood and there is also a need for large clinical trials. The Lab4b consortium of probiotics has been specifically designed to reflect the gut of new-born babies and infants [7,8]. A previous randomised control trial showed that Lab4b prevented atopic sensitisation to common food allergens and, hence, reduced the incidence of atopic eczema in early childhood [8]. In addition, Lab4b CM had neuroprotective effects in vitro [9]. The effects of Lab4 CM on key atherosclerosis-associated processes are not known and were investigated here on macrophages and VSMC, two key cell types implicated in the disease. Lab4b had many beneficial effects, including attenuation of chemokine-driven monocytic migration; macropinocytosis; oxLDL uptake; and proliferation of monocytes, macrophages, and VSMC together with PDGF-induced migration of VSMC. Lab4b CM also induced phagocytosis and cholesterol efflux from foam cells. In addition, Lab4b CM decreased the expression of key genes involved in the uptake of modified LDL and induced those implicated in the efflux of cholesterol from foam cells, thereby providing potential mechanisms for the anti-foam cell actions.

Our previous studies in relation to probiotics and atherosclerosis were focussed on Lab4P consortium that reduced plaque burden and content of lipids and macrophages, indicative of reduced inflammation, and increased the levels of plaque-stabilising smooth muscle cells in vivo [6]. In vitro studies used CM from Lab4 [*Lactobacillus acidophilus* CUL21 (NCIMB 30156) and CUL60 (NCIMB 30157), *Bifidobacterium bifidum* CUL20 (NCIMB 30153) and *Bifidobacterium animalis* subsp. *lactis* CUL34 (NCIMB 30172] or CUL66 (*Lactobacillus plantarum* CUL66) [6]. The effects of Lab4b CM seen here were similar to those of Lab4 except that Lab4b, but not Lab4, attenuated the expression of the LPL gene (Figure 4D). However, more differences were seen with CUL66. Thus, in contrast to Lab4b CM and Lab4 CM, CUL66 CM had no effect on the expression of ABCA1, ABCG1, and LXR-α, and attenuated the expression of LXR-β and ApoE [6]. In addition, CUL66 CM had no effect on PDGF-induced migration of smooth muscle cells [6]. These findings suggest potential strain-specific differences, and future studies should seek to delineate the effects of each bacterial species present in Lab4b or Lab4 on atherosclerosis in vitro and in vivo.

Not all effects of Lab4b CM were anti-atherogenic, as it increased the TBHP-induced ROS production (Figure 2) that could potentially increase oxidation of LDL. However, the role of ROS in atherosclerosis is complex and controversial, especially as many therapeutic approaches using antioxidants, such as vitamins C and E, in atherosclerosis have failed [1]. Focus has therefore shifted to modulators of oxidative stress, such as mitochondrial ROS, endothelial nitric oxide synthase, or major ROS-producing enzymes, such as NADPH oxidases [18,19]. Future studies should therefore investigate the effect of Lab4b on these modulators together with oxidation of LDL in vitro and in vivo.

The effect of probiotics on macrophage phagocytic activity has been investigated but the majority of these have not been in the context of atherosclerosis [20]. For example, increased phagocytosis was seen with a probiotic *Lactobacillus* strain and a novel strain of *Bacteroides fragilis* [21,22]. In the context of atherosclerosis, increased phagocytosis has been associated with decreased oxLDL uptake and reduced atherosclerotic plaque formation in ApoE^−/−^ mice [23]. Although phagocytic activity of macrophages is essential for the removal of apoptotic cells and debris from the plaques, it could potentially exert pro-atherogenic properties via phagocytosis of erythrocytes and platelets [24]. The significance of increased phagocytosis by Lab4b CM on foam cell formation and atherosclerosis therefore needs to be determined.

The impact of Lab4b CM on cell proliferation is of importance in the context of atherogenesis. Whilst macrophage proliferation is generally considered to be more important in advanced lesion development, monocytes play an essential role, particularly in early stages of the disease [25,26]. Studies have shown an association between decreased proliferation of macrophages and suppression of atherosclerosis and accelerated atherosclerosis development with increased proliferation of monocytes and macrophages in mouse model systems [25,26,27]. Our findings correlate well with a previous study that investigated the effects of specific probiotic strains on proliferation of peripheral blood mononuclear cells in vitro, showing a significant reduction in a concentration-dependent manner, similar to the anti-proliferative agent dexamethasone [28]. In the case of VSMC, whilst normal controlled proliferation can be beneficial, dysregulated proliferation contributes to plaque formation [29]. Our results therefore suggest that Lab4b CM has the potential to exert beneficial actions on dysregulated VSMC proliferation during atherosclerosis.

Lab4b CM also attenuated invasion of HASMC in vitro (Figure 6). Previous studies have shown that PDGF-BB and PDGF receptor β (PDGFRβ) are expressed in VSMC within atherosclerotic plaques, and their inhibition reduces their migration and proliferation and, subsequently, atherosclerotic lesion size [30,31,32]. For example, the tyrosine kinase inhibitor imatinib blocks the activity of the PDGF receptors, and diabetic ApoE^−/−^ mice treated with imatinib had reduced plaque areas that correlated with both reduced expression of PDGF-BB and phosphorylation of PDGFRβ [30,33]. It is possible that Lab4b CM inhibits pathological PDGF-induced invasion of VSMC via attenuation of PDGF receptor expression and/or activity, which therefore needs to be investigated.

Lab4b CM showed key anti-foam cell action by attenuating the uptake of lipoproteins and stimulating the efflux of cholesterol from foam cells (Figure 3). Analysis of gene expression revealed potential mechanisms of action. Thus, Lab4b CM attenuated the expression of two key scavenger receptors, SRA and CD36 (Figure 4), which have been shown to promote atherosclerosis in vitro and in vivo [34]. In addition, Lab4b CM, but not Lab4 CM or CUL66 CM [6], attenuated the expression of the LPL gene (Figure 4). LPL can associate with lipoproteins and increase the efficiency of their uptake by macrophages [35,36]. In addition, LPL also increases the production of pro-inflammatory cytokine tumour necrosis factor-α together with monocyte adhesion to endothelial cells and the proliferation of VSMC [36], all pro-atherogenic events. The Lab4b CM-mediated induction of cholesterol efflux was associated with increased expression of ABCA1, ABCG1, LXR-α, and LXR-β (Figure 4), suggesting increased cholesterol efflux, in part, via LXR and ABC transporter pathways [37]. LXR-α-mediated ABCA1/ABCG1-dependent macrophage cholesterol efflux has been associated with decreased lesion area and plaque lipid accumulation in mouse model systems [37,38]. In addition, it has been shown that the LXR activator TO901317 increased ABCA1/ABCG1 expression both in vitro and in vivo, thereby reducing foam cell formation in macrophages in vitro and atherosclerotic lesion area in vivo [37]. The role of ABCA1 in macrophage cholesterol efflux following LXR activation is well-established but the involvement of ABCG1 remains unclear [39]. It should be noted that changes in only mRNA expression were determined, as our previous studies have shown strong correlation between mRNA and protein expression of these genes [10,11]. However, future studies should also investigate changes in protein expression.

## 4. Conclusions

This study demonstrates, for the first time, that Lab4b CM has many beneficial effects on key atherosclerosis-associated processes in monocytes/macrophages and VSMC, including inhibition of proliferation and macrophage foam cell formation. This study also reveals the molecular mechanisms underlying such beneficial actions, such as decreased uptake of modified lipoproteins associated with reduced expression of genes such as CD36, SRA and LDL, and induced cholesterol efflux from foam cells associated with increased expression of LXR-ABC transporter genes. Future studies should therefore investigate the effects of Lab4b on atherosclerosis development in vivo in mouse model systems and in clinical trials.

## 5. Materials and Methods

### 5.1. Materials

All reagents were from Sigma-Aldrich (Poole, UK) or ThermoFisher Scientific (Paisley, UK) unless otherwise stated.

### 5.2. Preparation of Lab4b CM

Lab4b bacteria were cultured in de Man–Rogosa–Sharpe broth (0.1 mg/mL) for 18 h at 37 °C under low oxygen and non-agitating conditions (80% N_2_, 10% CO_2_, 10% O_2_, *v*/*v*) [6]. Following centrifugation at 1000× *g* for 10 min and washing with 1X PBS, the bacteria were resuspended in the same volume of tissue culture media and incubated as above for 5 h at 37 °C. The CM was collected following centrifugation, as above, adjusted to 7.4 using 0.5 M NaOH, supplemented with penicillin-streptomycin (both 10,000 U/mL), and filter-sterilised using 0.22-μm-pore filters [6]. Tissue culture media, treated similarly, was used as a vehicle control. The concentration of protein in each batch of CM was determined using the Pierce Coomassie Plus Assay Kit, to avoid batch specific variations associated with the use of set volume only, and stored in aliquots at −80 °C to avoid freeze/thaw cycles.

### 5.3. Cell Culture and In Vitro Assays

Culturing of THP-1 monocytes and their differentiation into macrophages was carried out as previously described [6,11,40,41]. Primary HASMC were cultured in smooth muscle cell growth medium, as described by the manufacturer (Sigma-Aldrich) [6]. Culturing of HMDM from buffy coats obtained from the National Blood Service, Wales, was performed, as in our previous study [6]. Each donor provided informed consent for use of human blood for non-transfusion purposes, and all experiments and associated ethical clearances were approved by Cardiff University.

Determination of cell viability by following the release of LDH into the medium, cell proliferation using crystal violet, rate of cell proliferation using the BrdU labelling and detection kit III, MCP-1-driven monocytic migration, uptake of Dil-labelled oxLDL, macropinocytosis using Lucifer yellow, phagocytosis using a Vybrant^®^ Phagocytosis Assay Kit, and radioactive-based cholesterol efflux from foam cells to ApoA1 acceptors were performed, as previously described [6,11,41,42,43,44]. ROS were measured using a 2′7′-dichlorofluorescin diacetate (DCFDA) Cellular ROS Detection Assay Kit (ab113851), according to the manufacturer’s instructions (Abcam), with TBHP (50 μM) used to produce ROS, as seen in pathological conditions [6,45]. PDGF-BB (designated as PDGF)-induced migration of HASMC was performed using a modified Boyden chamber with Matrigel-coated membrane, as in our previous studies [6,11].

### 5.4. Gene Expression

Purification of RNA, preparation of cDNA and RT-qPCR were carried out, as previously described, with transcript levels determined using the comparative Ct method [6,11,42,45]. The sequences of the primers were: 5′-AGCCATTTTAAAGATAGCTTTCC-3′ and 5′-AAGCTCTGGTTCTTATTCACA-3′ for CD36; 5′-GTCCAATAGGTCCTCCGGGT-3′ and 5′-CCCACCGACCAGTCGAAC-3′ for SRA; 5′-GAGATTTCTCTGTATGGCACC-3′ and 5′-CTGCAAATGAGACACTTTCTC-3′ for LPL; 5′-AGTGGAAACAGTTAATGACCAG-3′ and 5′-GCAGCTGACATGTTTGTCTTC-3′ for ABCA1; 5′-GGTGGACGAAGAAAGGATACAAGACC-3′ and 5′-ATGCCCGTCTCCCTGTATCCA-3′ for ABCG1; 5′-CCTTCAGAACCCACAGAGATCC-3′ and 5′-ACGCTGCATAGCTCGTTCC-3′ for LXR-α; 5-GCTAACAGCGGCTCAAGAACT-3′ and 5′-GGAGCGTTTGTTGCACTGC-3′ for LXR-β; 5′-CAGGAGCCGACTGGCCAATC-3′ and 5′-ACCTTGGCCTGGCATCCTG-3′ for ApoE; and 5-CTTTTGCGTCGCCAGCCGAG-3′ and 5′-GCCCAATACGACCAAATCCGTTGACT-3′ for GAPDH.

### 5.5. Statistical Analysis of Data

The number of independent experiments is indicated in the figure legends and each experiment was mostly performed in duplicate/triplicate. Values outside of 2 standard deviations of the mean were removed as outliers prior to statistical analysis. All normality of data, homogeneity of variance and statistical analysis was carried out using GraphPad Prism 9, and any transformation of data have been indicated in the figure legend. For two groups, an unpaired Student’s *t*-test or an unpaired Welch’s *t*-test were used for normally distributed data, and a Mann–Whitney test for not normally distributed data. Linear regression was also performed using GraphPad Prism 9. For more than two groups, a one-way ANOVA was used with Bonferroni or Dunnett post-hoc tests (significance defined as *p* ≤ 0.05).

## Figures and Tables

**Figure 1 ijms-24-03639-f001:**
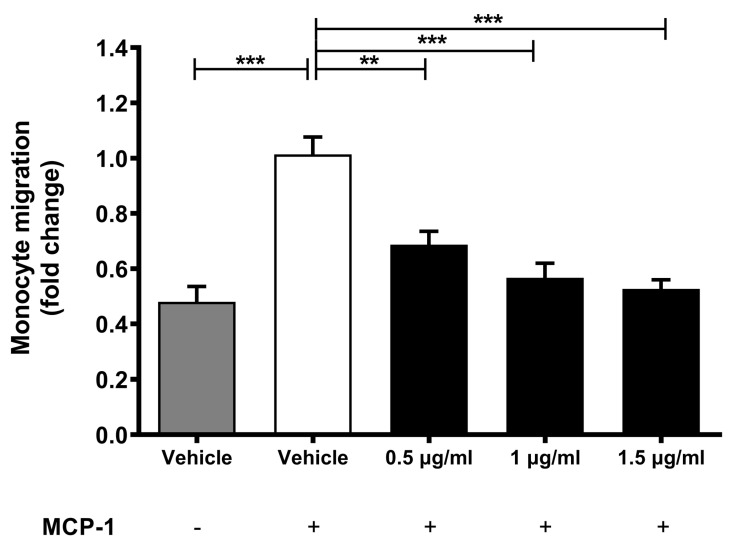
**Migration of human monocytes in response to three different concentrations of Lab4b CM.** MCP-1-induced migration of THP-1 monocytes was followed in the presence of either the vehicle or three different concentrations of Lab4b CM as shown (0.5 μg/mL, 1 μg/mL and 1.5 μg/mL). Cells treated with vehicle but in the absence of MCP-1 were also included for comparative purposes. Migration was determined as a percentage of total cells and shown as fold change in migration relative to the vehicle control with MCP-1 (20 ng/mL), which was arbitrarily assigned as 1. Data are mean ± SEM from five independent experiments and statistical analysis was carried out on log transformed data using one-way ANOVA with the Bonferroni post-hoc test (**, *p* ≤ 0.01; ***, *p* ≤ 0.001).

**Figure 2 ijms-24-03639-f002:**
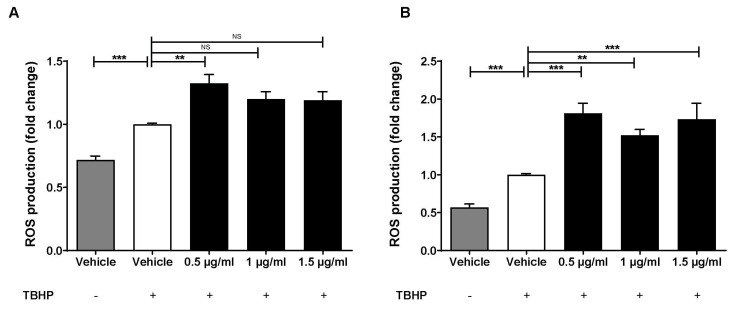
**Lab4b CM increases TBHP-induced ROS production in human monocytes and macrophages.** THP-1 monocytes (**A**) and macrophages (**B**) were treated with 50 μM TBHP and either vehicle or three different concentrations of Lab4b CM, as shown (0.5 μg/mL, 1 μg/mL and 1.5 μg/mL). Cells treated with vehicle but in the absence of TBHP were also included for comparative purposes. ROS production is shown as fold change to the vehicle control with TBHP, which was arbitrarily assigned as 1. Data are mean ± SEM from five independent experiments, and statistical analysis was carried out on sine–transformed (**A**) or log transformed (**B**) data using one-way ANOVA with the Bonferroni post-hoc test (**, *p* ≤ 0.01; ***, *p* ≤ 0.001; NS, not significant).

**Figure 3 ijms-24-03639-f003:**
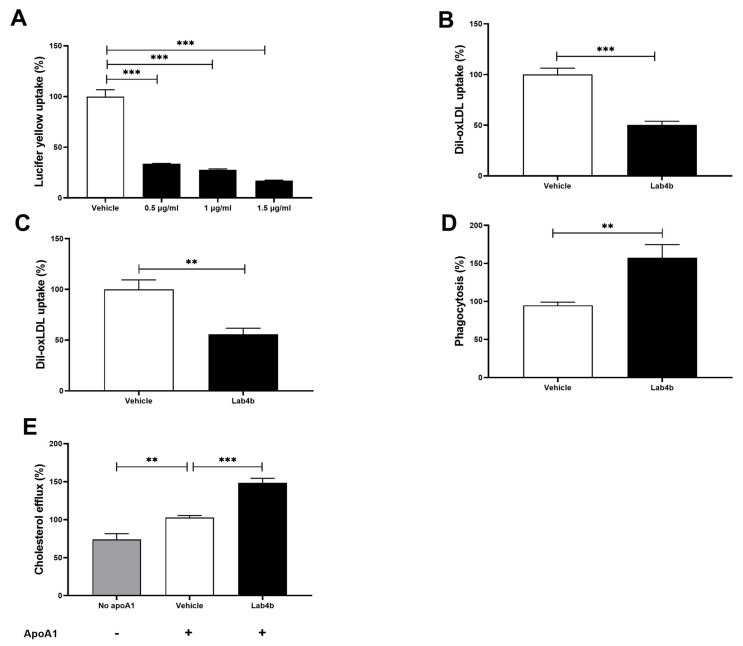
**Lab4b CM modulates several processes associated with foam cell formation in human macrophages.** THP-1 macrophages (**A**,**B**,**D**), HMDM (**C**) or THP-1 macrophage-derived foam cells (**E**) were used to determine macropinocytosis (**A**), Dil-oxLDL uptake (**B**,**C**), phagocytosis (**D**), and cholesterol efflux (**E**). Macropinocysosis was carried out using three different concentrations of Lab4b CM (0.5 μg/mL, 1 μg/mL, and 1.5 μg/mL), whereas, 1.5 μg/mL was used for all the other assays. The values in cells treated with vehicle alone (**A**–**D**) or vehicle and apolipoprotein (apo) A1 (10 μg/mL; (**E**) were arbitrarily assigned as 100%. Data are mean ± SEM from three independent experiments, and statistical analysis was carried out using one-way ANOVA with Dunnett (**A**; log transformed data) or Bonferroni (**E**) post-hoc test, an unpaired Student’s *t*-test (**B**,**C**) or unpaired Welch’s *t*-test (**D**) (**, *p* ≤ 0.01; ***, *p* ≤ 0.001).

**Figure 4 ijms-24-03639-f004:**
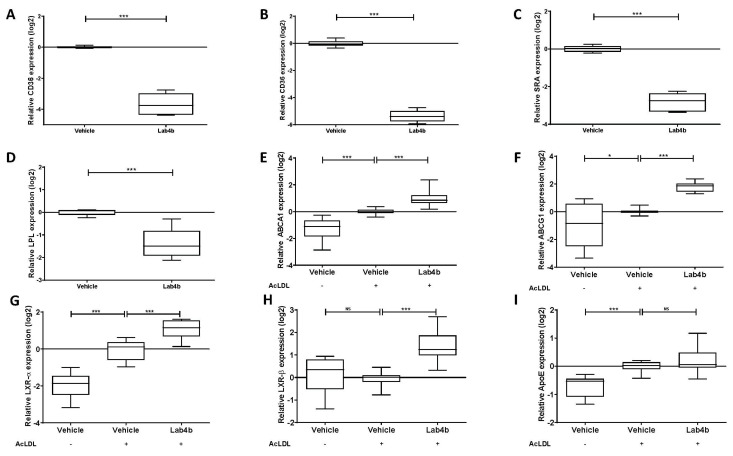
**Lab4b CM regulates the expression of several key genes implicated in the control of macrophage foam cell formation.** THP-1 macrophages (**A**,**C**,**D**), HMDM (**B**) or THP-1 macrophage-derived foam cells using acetylated LDL (AcLDL, 25 μg/mL; **E**–**I**) treatment for 24 h were incubated for 24 h with vehicle or Lab4b CM (1.5 μg/mL). For THP-1 macrophage-derived foam cells (**F**–**I**), cells treated with vehicle alone in the absence any AcLDL treatment were also included for comparative purposes. The expression of CD36 (**A**,**B**), SRA (**C**), LPL (**D**), ABCA1 (**E**), ABCG1 (**F**), LXR-α (**G**), LXR-β (**H**), and ApoE (**I**) was determined on purified RNA by real-time quantitative PCR (RT-qPCR), as described in Materials and Methods, and normalised to the glyceraldehyde 3-phosphate dehydrogenase (GAPDH) housekeeping gene. The data are presented as box-and-whisker plots of log2 fold-change in gene expression relative to the vehicle control, where whiskers represent minimum-to-maximum fold-change from three independent experiments. Statistical analysis was performed using an unpaired Welch’s *t*-test (**A**,**C**,**D**), an unpaired Student’s *t*-test (**B**) or one-way ANOVA with the Bonferroni post-hoc test (**E–I**) (*, *p* ≤ 0.05; ***, *p* ≤ 0.001; NS, not significant).

**Figure 5 ijms-24-03639-f005:**
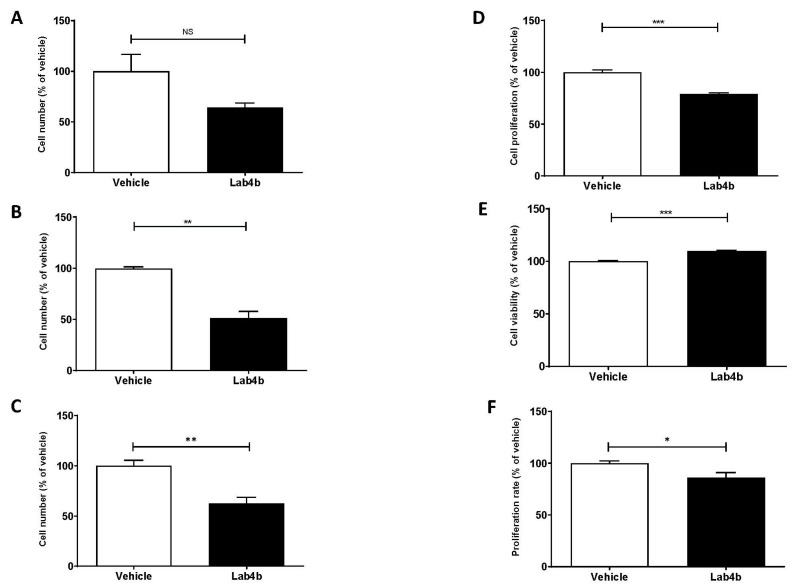
**Lab4b CM attenuates proliferation of human monocytes and macrophages.** (**A**–**C**), THP-1 monocytes were treated with either vehicle or 1.5 μg/mL Lab4b CM and cell numbers counted at days 2, 5, and 7 and represented to the vehicle control, which was arbitrarily assigned as 100%. (**D**), THP-1 macrophages were treated for 48 h with vehicle or 1.5 μg/mL Lab4b CM and change in proliferation, as assessed by the crystal violet assay, was determined as a percentage relative to the vehicle control, which was arbitrarily assigned as 100%. (**E**), The supernatant from the same cells was used to assess viability via LDH activity and determined as a percentage of the vehicle control, which was arbitrarily assigned as 100%. (**F**), The proliferation rate of THP-1 macrophages was determined by following BrdU incorporation following treatment of THP-1 macrophages with vehicle or 1.5 μg/mL Lab4b CM. Proliferation rate was determined as a percentage of vehicle, which was assigned as 100%. Data are mean ± SEM from three independent experiments, and statistical analysis was carried out using an unpaired Student’s *t*-test (*, *p* ≤ 0.05; **, *p* ≤ 0.01; ***, *p* ≤ 0.001; NS, not significant).

**Figure 6 ijms-24-03639-f006:**
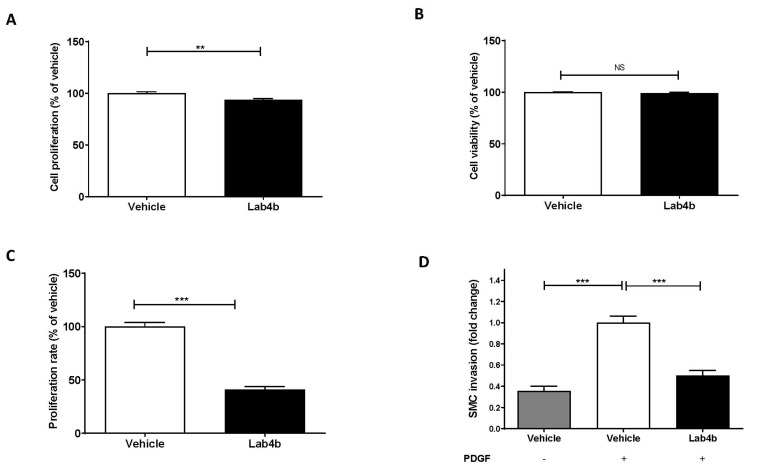
**Lab4b CM affects proliferation and migration of HASMC.** (**A**,**B**), HASMC were treated for 7 days with vehicle or 1.5 μg/mL Lab4b CM and cell proliferation was determined using crystal violet. The supernatant from the same cells was used to determine cell viability by following LDH activity. (**C**), HASMC were treated with vehicle or 1.5 μg/mL of Lab4b CM for 2 days and the rate of proliferation was determined by BrdU incorporation. The values from Lab4b CM-treated cells are represented in relation to the vehicle control, which was arbitrarily assigned as 100%. (**D**), Invasion of HASMC in response to PDGF (20 ng/mL) was determined following treatment with vehicle or 1.5 μg/mL of Lab4b CM for 4h. Cells incubated with vehicle in the absence of PDGF were also included for comparative purposes. The number of migrated cells were counted and averaged per five high-powered field and represented as fold-change to the vehicle control with PDGF, which was arbitrarily assigned as 1. In all cases, data are mean ± SEM from three independent experiments with statistical analysis carried out using an unpaired Student’s *t*-test (**A**–**C**) or one-way ANOVA with the Bonferroni post-hoc test (**D**) (**, *p* ≤ 0.01; ***, *p* ≤ 0.001; NS, not significant).

## Data Availability

The data presented in this study are available on request from the corresponding author.

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
