# Peer review of "Anti-Atherogenic Actions of the Lab4b Consortium of Probiotics In Vitro"

_ijms, 2023, doi:10.3390/ijms24043639_

Round 1

Reviewer 1 Report

O’Morain and colleagues presented an original research article aimed at evaluating the in vitro anti-atherogenic effects of probiotics. For this purpose, they performed in vitro experiments on monocyte/macrophage and heart cell lines by using cell culture medium conditioned by the Lab4b consortium. Overall, the research idea is interesting, however, the experimental design is very simple and some controls and experiments are missing. Below are reported some minor/major comments which will improve the quality of the manuscript:

1) In the Results section, the authors state: “The highest concentration of Lab4b CM used in our studies was 1.5 ug/ml so the effect of this concentration on cell viability was determined.”. How did the authors select this concentration? You have to show data on IC50 assays at different concentrations evaluating how different concentrations of Lab4b CM influences cell viability and cell doubling time. Please address this issue;

2) In Figure 1 there are missing controls. Indeed, the authors have to evaluate also the migration rates of Vehicle (-) and the three different concentrations of Lab4b CM. Same comment for Figure 2;

3) The authors evaluated the expression levels of different genes. What about the protein expression of such genes? Did you have some ELISA or WB data? You should consider to add such experiments to further corroborate your data;

4) At the beginning of the Discussion section, it would be useful to add a more general description of the benefits of using probiotics in different pathologies. Currently, there are a lot of doubts about the real effects of probiotics on human health. Therefore, you should add a brief description describing the benefits of using probiotics in different pathologies, including cancer, cardiovascular diseases and neurodegenerative disorders. For this purpose, please see:

- https://doi.org/10.3892/wasj.2019.13

- https://doi.org/10.3390/nu13082878

- https://doi.org/10.3389/fnagi.2022.730036

- https://doi.org/10.1039/C5FO01190F

5) What concentration of conditioned medium was used (volume/volume)? Did the authors evaluate the IC50 of the conditioned medium? Please, clarify this in the Methods section.

Author Response

We thank both reviewers for their constructive comments on this manuscript that has resulted in a mark improvement of the revised version. We note the positive tone that has been adopted (“the research idea is interesting” -reviewer 1; “the overall manuscript is well-written and designed” -reviewer 2). We wish to respond to the other comments that have been raised using track changes where required.

Response to Reviewer 1:

Point 1:  In the Results section, the authors state: “The highest concentration of Lab4b CM used in our studies was 1.5 ug/ml so the effect of this concentration on cell viability was determined.”. How did the authors select this concentration? You have to show data on IC50 assays at different concentrations evaluating how different concentrations of Lab4b CM influences cell viability and cell doubling time. Please address this issue.

Response: As in our previous study (reference 6), three different concentrations of probiotic CM were used (0.5 μg/ml, 1 μg/ml and 1.5 μg/ml in this manuscript -standard increase) with the method for determining concentration included in Materials and Methods ("the concentration of protein in each batch of CM was determined using the Pierce Coomassie Plus Assay Kit"). As the final protein concentrations between batches varied, the minimum concentration from different batches (1.5 μg/ml) was used as the highest concentration in dose response experiments (0.5 μg/ml, 1 μg/ml and 1.5 μg/ml -standard increase). The intention was not to investigate which concentration produced 50% reduction in cell viability but to confirm that no inhibition of this was seen so the results that are obtained with atherosclerosis-based parameters are not due to compromised cell viability but because of the action of CM. We feel that this was sufficiently clear in the manuscript (i.e. "No detrimental reduction in cell viability by Lab4b CM was seen in THP-1 macrophages, HMDM and HASMC (Fig. S1))". Please note that many studies use set volume of CM, thereby not taking batch specific variations into account, but we have removed this potential source of variation by using a set concentration. We have now incorporated "to avoid batch specific variations with use of set volume only......" in Materials and Methods to add clarity (lines 381-382). 

Point 2: In Figure 1 there are missing controls. Indeed, the authors have to evaluate also the migration rates of Vehicle (-) and the three different concentrations of Lab4b CM. Same comment for Figure 2.

Response: The approach used here is based on our previous published research (references 6, 11, 37, 38) where the focus is on the effect of probiotic CM on MCP-1 driven monocyte migration that is relevant to atherosclerosis and not constitutive migration seen in the absence of the chemokine. Thus, the left two histograms show that MCP-1 induces the migration of THP-1 monocytes and this MCP-1-induced migration, which is relevant to atherosclerosis, is attenuated by Lab4b CM. The same point applies to Figure 2 where we are investigating TBHP-induced ROS production and not constitutive ROS production that tends to be negligible in the absence of a stimulus. In light of this comment, we have revised the text slightly so that this point is clear to the reader (lines 89-90 and lines 120-121.

Point 3: The authors evaluated the expression levels of different genes. What about the protein expression of such genes? Did you have some ELISA or WB data? You should consider to add such experiments to further corroborate your data.

Response: Our previous studies (e.g. references 10, 11 and several others not included in references) have shown that changes in mRNA for these genes are accompanied by changes in the expression of the corresponding proteins. Our studies here were therefore restricted to mRNA expression. We have therefore elaborated on this point on lines 356-358.

Point 4: At the beginning of the Discussion section, it would be useful to add a more general description of the benefits of using probiotics in different pathologies. Currently, there are a lot of doubts about the real effects of probiotics on human health. Therefore, you should add a brief description describing the benefits of using probiotics in different pathologies, including cancer, cardiovascular diseases and neurodegenerative disorders. For this purpose, please see:

- https://doi.org/10.3892/wasj.2019.13

- https://doi.org/10.3390/nu13082878

- https://doi.org/10.3389/fnagi.2022.730036

- https://doi.org/10.1039/C5FO01190F

Response: We thank the reviewer for this comment and have incorporated some text at the start of the discussion (lines 234-237). The subsequent references have been renumbered accordingly.

Point 5: What concentration of conditioned medium was used (volume/volume)? Did the authors evaluate the IC50 of the conditioned medium? Please, clarify this in the Methods section.

Response: We have already commented on this in relation to point 1 with appropriate changes in the manuscript. Thus, we were not investigating viability/cell death by the CM by determining IC50 but ensuring that no detrimental effect was seen so changes in atherosclerosis-assiciated parameters are not because of reduced cell viability. Different concentrations of CM was used (same volume throughout) and the concentrations are indicated in the text.

We look forward to the outcome of this resubmission.

Best Wishes,

Dipak

Reviewer 2 Report

 Anti-atherogenic actions of the Lab4b consortium of probiotics in vitro

 O’Morain et al.,

The overall manuscript is well-written and designed.  I congratulate the O’Morain all the time for the study.

I’m sending my considerations for improving your manuscript.

Title

Anti-atherogenic actions of the Lab4b consortium of probiotics: an in-vitro study

Abstract

Introduction

For an elegant paper, please, include one final phrase describing the study objective.

Results

Please, do not include references in the Results section.

The result is  Result.

Add the references and themes as references in the Materials and Methods section.

Lab4b CM has no detrimental effect on the cell viability of human macrophages and VSMC

Line 77-81. Descriptions are Materials and Methods section.

Lab4b attenuates chemokine-driven monocytic migration

Line 86-87. Descriptions are Materials and Methods section.

Lab4b CM has beneficial actions on several parameters associated with macrophage foam cell formation

Line 102-105. Descriptions are Materials and Methods section.

Figure 3E. In Materials and Methods section was not describe the ApoA 1.

The effect of Lab4b CM on monocyte/macrophage proliferation.

Excluded dote of subheading

Line 162-163. Descriptions are Materials and Methods section

Lab4b CM attenuates HASMC proliferation and PDGF-induced migration

Line 194. “Linear regression analysis showed that…” Analysis was not described as a Linear regression statistics approach in the STATISTICAL ANALYSIS OF DATA section.

Please, refer to this approach in order to, identify adequate results.

Additionally:

Please, add Supplementary data describing a Sensitivity Analysis as all samples that were removed as outliers and analysis included outliers.

FIGURES:

My old eyes have difficulty identifying small details in figures. Please, increase the figure’s resolution and size for a better image of your results.

Materials and Methods

Preparation of Lab4b CM.
Describe the threshold accepted of protein concentration for freezing the cultured Lab4b bacteria.

Please, describe um subheading
Human and experimental ethics
For human experiments and cell donation… Each donor provided informed consent and all experiments were approved by Cardiff University.

Was full protocol submitted to Ethics regulatory Committee? Describe.

Gene Expression

Please, include the data of intra-test variance of the PCR results.

Statistical analysis of data

Descriptions of a Sensitivity Analysis as all samples that were removed as outliers and analysis included outliers applied the statistical adjustments necessary.

Discussion

I’m very fascinated by contextualization based on previous findings and hard work from your group on the research theme. But I things, in this present paper, the discussion could be initiated with our findings of the study.

Author Response

We thank both reviewers for their constructive comments on this manuscript that has resulted in a mark improvement of the revised version. We note the positive tone that has been adopted (“the research idea is interesting” -reviewer 1; “the overall manuscript is well-written and designed” -reviewer 2). We wish to respond to the other comments that have been raised using track changes where required.

Point 1: Title: Anti-atherogenic actions of the Lab4b consortium of probiotics: an in-vitro study

Response: We have retained the title as all the above points are already present in the current title ("Anti-atherogenic actions of the Lab4b consortium of probiotics in vitro")

Point 2: "Introduction For an elegant paper, please, include one final phrase describing the study objective".

Response: Done (lines 68-70).

Point 3: Please, do not include references in the Results section. The result is  Result. Add the references and themes as references in the Materials and Methods section.

Response: The references were included as additional justification of model (e.g., THP-1 macrophages, HASMC) or assays (e.g., oxidised LDL uptake, macropinocytosis; proliferation of cells) used. These references are not for methods but justification to readers on why these models and approaches were used. This adds clarity and makes it easy for the reader to follow the results, particularly a non-specialist in the field. References are therefore retained. This approach has been used in all over 90 publications from my laboratory and many readers have complemented it in terms of helping them to follow the results.

Point 4:  Lab4b CM has no detrimental effect on the cell viability of human macrophages and VSMC. Line 77-81. Descriptions are Materials and Methods section.

Response: This point has been addressed above (point 3) and the references are not description of Materials and Methods, which are provided at end, but justification on use of THP-1 macrophages, HASMC and MCP-1 to help the reader, particularly a non-specialist, to follow the results.

Point 5: Lab4b attenuates chemokine-driven monocytic migration. Line 86-87. Descriptions are Materials and Methods section.

Response: We have retained reference 3 and associated description as this relates to justification of use of MCP-1. However, we have removed "determined using a modified Boyden chamber" as this is indeed methods.

Point 6: Lab4b CM has beneficial actions on several parameters associated with macrophage foam cell formation Line 102-105. Descriptions are Materials and Methods section.

Response: We have retained the description as it's not methods but helping the reader, particularly a non-specialist, on what steps are involved in macrophage foam cell formation and why they were used in the study.

Point 7: Figure 3E. In Materials and Methods section was not describe the ApoA 1.

Response: We had stated that "....cholesterol efflux from foam cells were performed as previously described" citing appropriate references but have now added ApoA1 to the description (line 393).

Point 8: The effect of Lab4b CM on monocyte/macrophage proliferation. Excluded dote of subheading

Response: Done

Point 9: Line 162-163. Descriptions are Materials and Methods section

Response: As described above, the reference is not Materials and Methods but to justify, and thereby help the reader, why proliferation was assessed as this is increasingly been found to play an important role in atherosclerosis.

Point 10:  Lab4b CM attenuates HASMC proliferation and PDGF-induced migration. Line 194. “Linear regression analysis showed that…” Analysis was not described as a Linear regression statistics approach in the STATISTICAL ANALYSIS OF DATA section. Please, refer to this approach in order to, identify adequate results.

Response: This information was present in Fig. S3 but now has been added in Fig. S2 and the text on statistical analysis of data  (lines 601-602).  

Point 11: Additionally: Please, add Supplementary data describing a Sensitivity Analysis as all samples that were removed as outliers and analysis included outliers.

Response: Each experiment was mostly carried out in duplicate or triplicate so removal of any outliers relate to a technical repeat. The number of independent experiments used for statistical analysis are as indicated in each figure legends. We have now added the point that each experiment was mostly performed in duplicate or triplicate on lines 598-599 of the revised version.

Point 12:  FIGURES: My old eyes have difficulty identifying small details in figures. Please, increase the figure’s resolution and size for a better image of your results.

Response: Our apologies but the formatting was by the journal that requires pasting of figures where they are supposed to appear. We have tried to increase the size of the images where possible and hope that it's more easier to read.

Point 13: Materials and Methods Preparation of Lab4b CM.
Describe the threshold accepted of protein concentration for freezing the cultured Lab4b bacteria.

Response: The concentrations of proteins obtained was standard (minimum in different batch of 1.5 μg/ml) so there are no issues regarding freezing. The most important point is that the CM was stored in aliquots to avoid freeze-thaw cycles that could cause issues and this has been mentioned in the text.

Point 14: Please, describe um subheading. Human and experimental ethics
For human experiments and cell donation… Each donor provided informed consent and all experiments were approved by Cardiff University. Was full protocol submitted to Ethics regulatory Committee? Describe.

Response: We have been using HMDM from monocytes isolated from buffy coats for many studies. The reader was directed to reference 6 as our recent protocol using M-CSF but also includes references 10 and 11. The description is what we have used in all such publications. However, we have added additional text in the revised manuscript to add more clarity on this point (lines 388-389).

Point 15: Gene Expression. Please, include the data of intra-test variance of the PCR results.

Response: The intra-test variance in all our PCR in this and other studies are very low and as all experiments are carried out in triplicate, occasional outlier if present are removed as indicated in response to point 11.

Point 16: Statistical analysis of data. Descriptions of a Sensitivity Analysis as all samples that were removed as outliers and analysis included outliers applied the statistical adjustments necessary.

Response: This is addressed above (point 11).

We look forward to the outcome of this resubmission.

Best Wishes,

Dipak